# Deep Learning for interpretable end-to-end survival prediction in gastrointestinal cancer histopathology

Narmin Ghaffari Laleh (1), Amelie Echle (1), Hannah Sophie Muti (1),

Katherine Jane Hewitt (1), Volkmar Schulz (2,3,4,5), Jakob Nikolas Kather (1,6,7)

(1) Department of Medicine III, University Hospital RWTH Aachen, Aachen, Germany

(2) Department of Physics of Molecular Imaging Systems, Experimental Molecular Imaging, RWTH Aachen University, Aachen, Germany

(3) Fraunhofer Institute for Digital Medicine MEVIS, Bremen, Germany

(4) Comprehensive Diagnostic Center Aachen (CDCA, University Hospital Aachen, Aachen, Germany

(5) Hyperion Hybrid Imaging Systems GmbH, Aachen, Germany

(6) Medical Oncology, National Center for Tumor Diseases, University Hospital Heidelberg, Heidelberg, Germany

(7) Pathology & Data Analytics, Leeds Institute of Medical Research at St James's, University of Leeds, Leeds, UK

## Abstract

Digitized histopathology slides contain a wealth of information, only a fraction of which is being used in clinical routine. Deep learning can extract subtle visual features from digitized slides and thus can infer clinically relevant endpoints from raw image data. While classification and regression methods are well established in this domain, end-to-end prediction of patient survival still remains a comparably novel approach. To account for different follow-up times and censored data, previous approaches have largely used discretized survival data. Here, we demonstrate and validate EE-Surv, a powerful yet algorithmically simple method to predict survival directly from whole slide images which we validate in colorectal and gastric cancer, two clinically relevant and markedly different tumor types. We experimentally show that our method yields a highly significant prediction of survival and enables explainability of predictions. Our method is publicly available under an open-source license and can be applied to any type of disease.

## Introduction

For virtually every patient with a malignant tumor, histopathological tissue slides stained with hematoxylin and eosin (H&E) are available. These slides are increasingly being digitized in clinical routine, yielding gigapixel images which are accessible for computational analysis. In recent years, Deep Learning applications on histopathology images resulted in a high performance in classical tasks such as tissue segmentation, object detection and quality control. In addition Deep Learning has been used for more challenging "end-to-end" tasks such as disease subtyping and mutation detection [1–5]. In particular, end-to-end prognostication of survival is of high clinical relevance in treatment selection and follow-up of cancer patients.

Unlike for simple classification tasks, there is no standard method available for prediction of survival from histopathology images. Some previous studies have used Deep Learning for tissue segmentation and using the results to fit survival prediction models [6]; other studies have used Deep Learning to predict discretized survival from whole slide images [7] and some recent studies have aimed at specific tumor types (colorectal cancer [8], brain cancer [9], liver cancer [10] and mesothelioma [11]). Additionally, most of the proposed algorithms for survival prediction from histopathological images utilise a high number of preprocessing steps, like clustering the extracted tiles [12,13], generating regions of interest (ROI) [14]. However, to date, these approaches remain insular and there is no validated consensus method for survival prediction from raw histology slides. In any survival analysis, there are two main quantities. The survival Function $S(t)$, which is the probability of survival beyond time $t$ and the hazard function $h(t)$ which is the probability of an event occurring in the time interval. Hazard function consists of two main parts; the baseline hazard function and the risk function. So in general, while the survival function describes the absence of an interested event, the hazard function indicates the occurrence of the event.

In this study, we aimed to develop and validate a simple, versatile and efficient method for survival prediction directly from histopathology images. We present EE-Surv and applied this method to colorectal and gastric cancer, two clinically relevant but markedly distinct tumor types. We demonstrate a high end-to-end prediction performance as well as explainability and algorithmic efficiency of our method which can be applied to any tumor type.

## Methods

### Data Sets

In this study, we used digitized diagnostic whole slide images of two cohorts (N=413 patients with colorectal cancer, TCGA-CRC [15] and N=362 patients with gastric cancer, TCGA-STAD [16]) from The Cancer Genome Atlas Program (TCGA). Supplementary figure 1 shows the summary of these data sets. We excluded all patients for which survival data or slides were not available. Each patient in these cohorts has a record of time and an event indicator ($\delta \in \{0\,;\,1\}$, in the following time, 0 : event did not happen, 1: event happend). In both cohorts, we evaluate the predictive performance of EE-Surve by three-fold patient-level cross-validation, ensuring that no data from a patient in the training set was ever part of the test set in the same cross-validation run.

### EE-Surv

EE-Surv is an End-To-End deep learning model to predict survival directly from histopathology whole slide images (WSIs) with a minimum amount of pre- and post processing. Figure 1 illustrates the general workflow of EE-Surv. The pre- and post-processing are a standard approach in the field which has been previously used in classification problems [5], thereby keeping EE-Surv as simple as possible. All source codes for preprocessing are available at https://github.com/KatherLab/preProcessing and all source codes for EE-Surv are available at https://github.com/KatherLab/Survival .

### Image pre-processing

Due to the large size of WSIs, it has been discussed in previous studies that tessellation of WSIs and generating smaller tiles is a useful initial step [17–19]. In our EE-Surv model the input is normalised; smaller tiles of 512×512×3 are resized to the shape required by our model for training. Tiles are extracted from the whole slides without using any manual annotations. Normalization of the extracted tiles reduces the possibility of having bias among patients from different studies and/or slide-readers [20,21]. Specifically, we used the Macenko method [22] which converts the RGB color vector to its corresponding optical density (OD) values and uses these values to extract the metrics of the stain vector and the saturation of the stains [23].

**Model training**

Transfer learning is an established solution to save computational time and power and its high performance in histopathology has been shown in various studies [24]. Here, we used a ResNet-50 [25] which is pre-trained on ImageNet [26]. The original output layer has been replaced by a layer with single output and the linear activation function. Since the number of extracted tiles per slide varies among patients, we randomly selected 250 tiles per WSI and assigned the following time and event of the WSI to each tile. We split each cohort into 3 parts and using k-fold cross validation techniques evaluated the performance of the designed model [27]. The most important part of training EE-Surv is the Cox proportional Hazard loss function which we used to optimize the parameters of network while training. The fully connected layer of the modified network results in a risk prediction for each input image. This risk is the product of the layer weights $(1024 \times 1)^T$ and the inputs to this layer $(1024 \times 1)$. These risks are used in Cox proportional hazards layer to minimize the negative partial log likelihood and via backpropagation optimize the model weights, biases and the convolutional kernels [9].

**Post-processing and statistical analysis**

After training the model and generating the risk scores for each tile, we aggregate the scores to generate one risk score per patient. For statistical analysis, we use patient-level scores and split the patients at the median, generating a high-risk and a low-risk group. We use Kaplan-Meier curves to visualize survival differences, test statistical significance with a log-rank test and with a univariate and multivariate Cox proportional hazard model, the latter including tumor stage and age. In addition, we used the tile-wise prediction scores to generate slide-level heatmaps and selected high scoring tiles for a reader study.

## Results

**Deep Learning can predict survival in colorectal cancer**

We trained EE-Surv to predict survival in a multicentric cohort of colorectal cancer patients (TCGA-CRC) in a cross-validated way. When stratifying the patient prediction scores at the median, we found that high predicted risk scores corresponded to a poor survival (Figure 2a) with a highly statistically significant difference between high and low scoring patients (log rank p-value = 0.0021). In addition, we fitted a univariate Cox proportional hazard model, demonstrating a hazard ratio (HR) of 0.5038 (0.3227, 0.7864) for prediction of death by patients

with a low predicted risk score. This was again highly significant (p = 0.00255, Suppl. Table 1a). To rule out confounding factors we combined the EE-Surv based predicted risk score with two powerful conventional risk factors, patient age and tumor stage. Again, this multivariable model showed that EE-Surv can significantly predict risk of death (p = 0.00265, Suppl. Table 1b)

**Deep Learning can predict survival in gastric cancer**

Compared to colorectal cancer, gastric cancer can have a much more heterogeneous histological appearance. We assessed the prognostic performance of EE-Surv in a large cohort of gastric cancer patients (TCGA-STAD). Again, we found that a high predicted risk score was associated with a significantly shorter survival (log rank p-value = 0.0074), which was reproducible in a univariate Cox proportional hazard model (HR for death in low-scoring patients of 0.635 [0.4541, 0.888], p = 0.00795, Suppl. Table 2a). In a multivariable Cox proportional hazard model that included EE-Surv score, age and tumor stage, the risk prediction by age and stage were highly significant (p < 0.001 for either, Suppl. Table 2b). However, while EE-Surv reached a HR of 0.7342 (0.5093, 1.058), this effect was found not to be significant (p = 0.0978) in multivariable analysis.

## Discussion

In this study, we presented and evaluated EE-Surv, an algorithmically simple yet powerful end-to-end risk prediction tool for digital pathology. Unlike some previous methods, EE-Surv does not require dichotomization of outcomes and includes censored patients in training by using a Cox proportional hazard model loss function. Although we use a resnet50 model as the backbone of EE-Surv, other deep convolutional neural networks and other architectures such as vision transformers can be used with EE-Surv. All of our source codes are publicly available, allowing reproduction and extension of our methods. Crucially, we demonstrate the plausibility of EE-Surv-based predictions not only by statistical models, but also by a blinded user study involving a pathologist. We show that without being explicitly trained on image features with known association to risk of death, EE-Surv learns to detect these features. One of these features is infiltration of tumors by lymphocytes, which has been demonstrated to be of prognostic relevance in colorectal cancer more than a decade ago. [28] Future studies should focus on external validation of our findings in additional patient cohorts and other clinical scenarios. Furthermore, before real-world use of our methods, clinical trials evaluating the usage and the clinical consequences of our proposed algorithm are required. From a technical

standpoint, different aggregation methods for pooling tile-level predictions on the level of patients could conceivably further boost performance, although simple aggregation functions have been shown to perform on par with highly parametrized models. [29] Finally, the necessity of tesselating gigapixel images in histopathology into smaller image tiles is due to the memory limitation of graphics processing unit (GPU) memory. The broad availability of GPU devices with enough memory to train directly on WSI could eliminate this preprocessing step in the future.

**Figures**

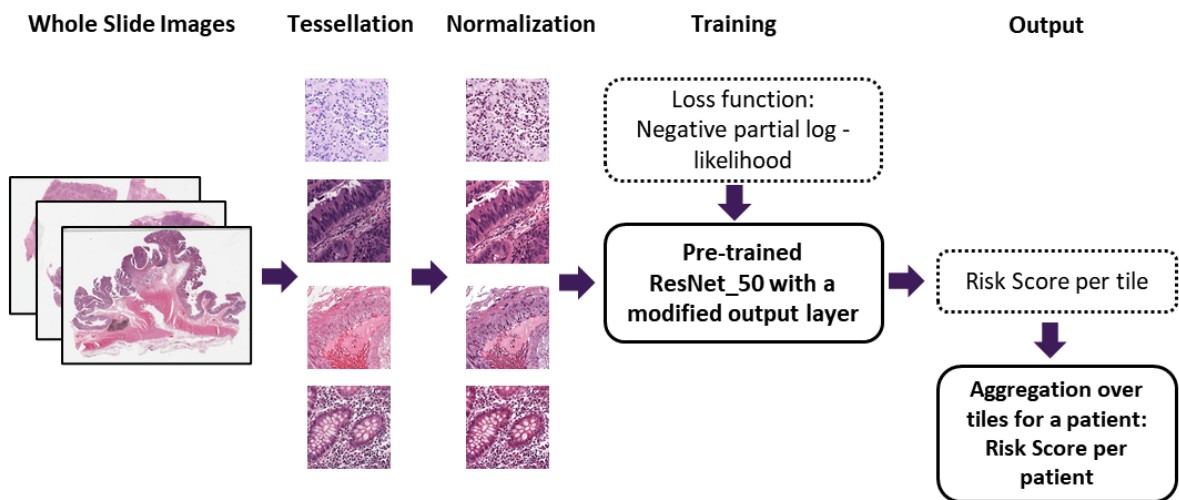

**Figure 1** General Workflow of EE-Surv. The simple workflow of EE-Surv starts with tessellation of the whole slide images into smaller tiles. Then the extracted tiles are normalized to have the same color distribution to remove the possible biases. The modified pretrained ResNet-50 is used to train the network and this will result in a risk score per tile. The average risk score over the all tiles selected per patient, is used as a final risk score for each patient.

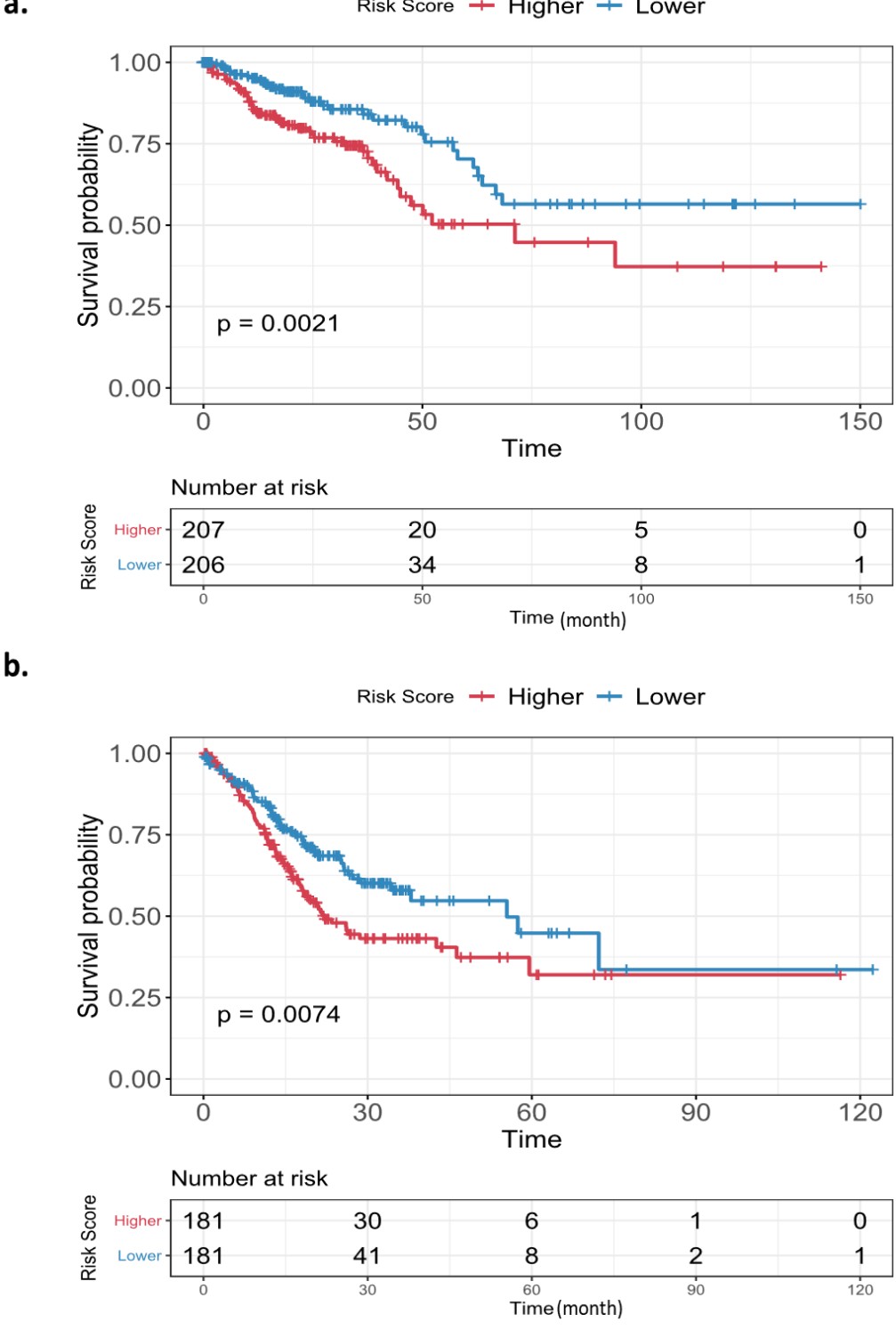

**Figure 2** Kaplan Meier plots for a) TCGA-CRC b) TCGA-STAD. We calculated the median of generated risk scores for each cohort and based on the median value, splitted the patients into higher and lower risk groups.

**Supplementary Figures**

a.

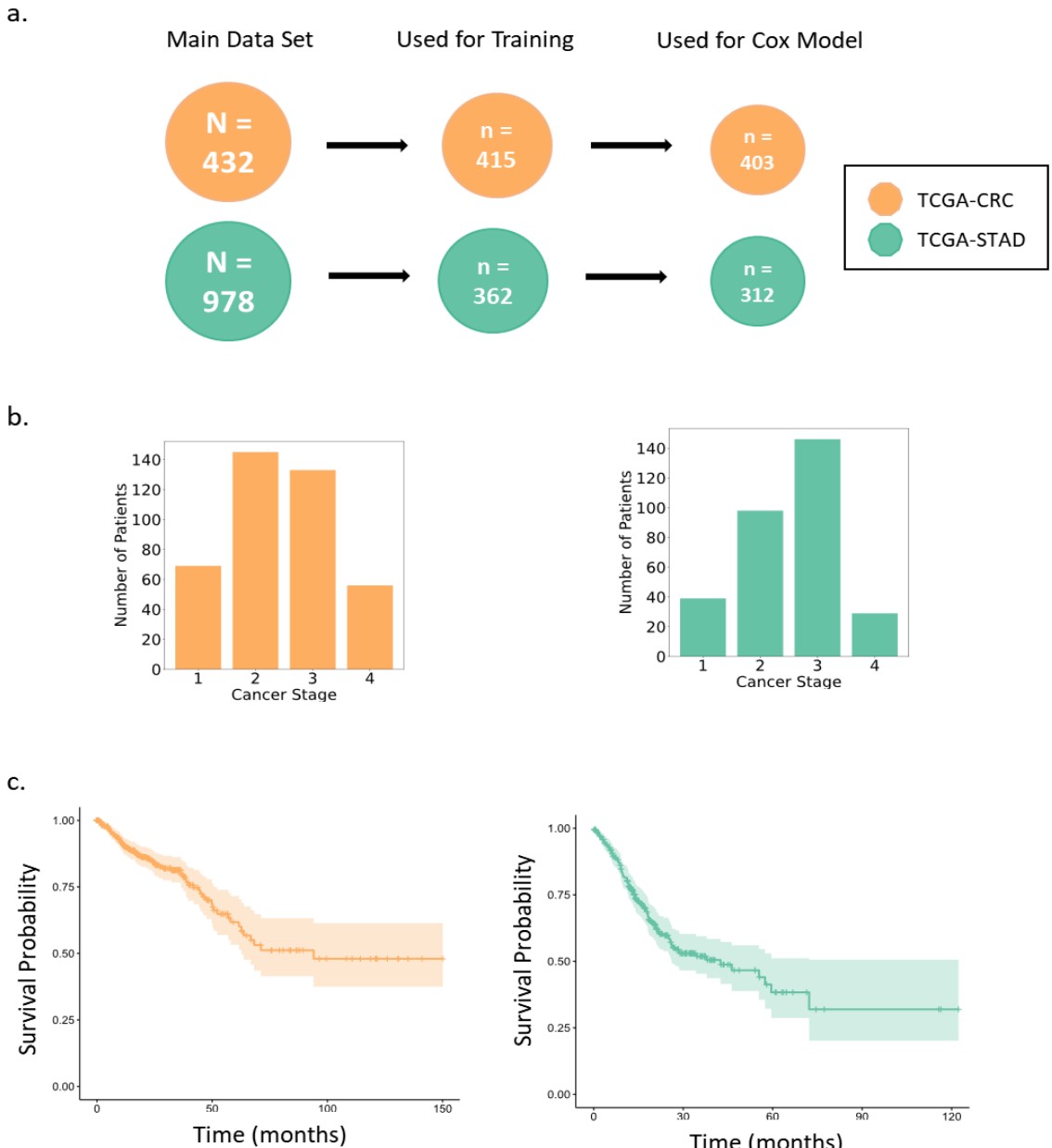

**Supplementary Figure 1** Data set Description. a) shows the number of the patients in the original data set, number of patients, who has all information required for the training (tiles, time and event of interest) and finally number of patients which contain all required data for multivariate cox regression (age and stage of cancer). b) Histogram of the final data set, for the stage of cancer. c) shows the general survival plot for both cohorts.

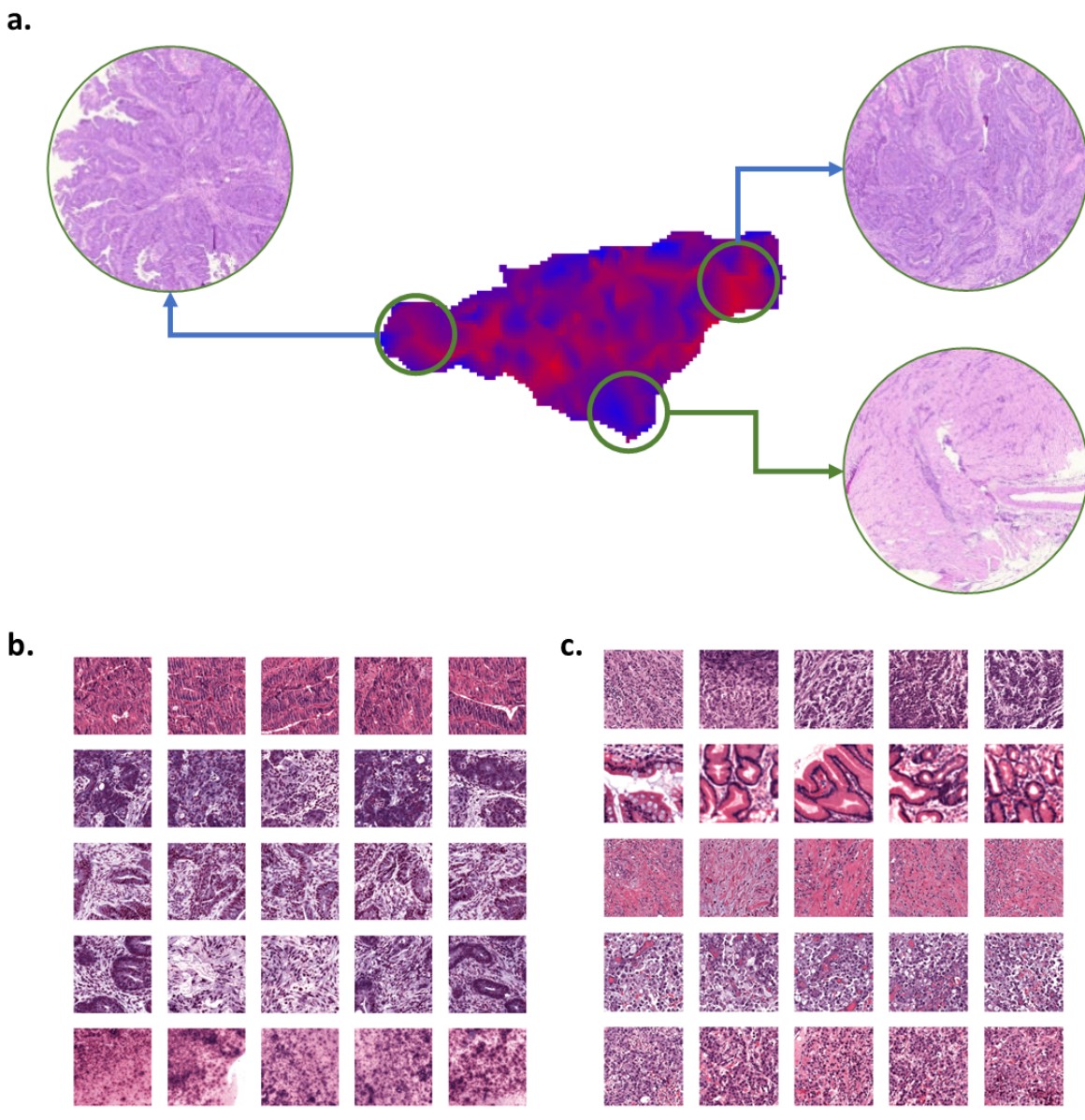

**Supplementary Figure 2** Explainability of the EE-Surv. a) An example heatmap from the TCGA-CRC cohort. In this heatmap, the red color correlated with the high risk score value and the blue color shows the low risk score value. b) Shows the 5 high score tiles for the first 5 high score patients for TCGA-CRC. c) Shows the 5 high score tiles for the first 5 high score patients for TCGA-STAD.

**Supplementary Tables**

**a.**

| coxph(formula = Surv(time, event) ~ groups) | | | | | | |
|---|---|---|---|---|---|---|
| | Coeff | Exp (Coeff) | Lower 0.95 | Upper 0.95 | z | Pr (>\|z\|) |
| Groups (Lower) | -0.6856 | 0.5038 | 0.3227 | 0.7864 | -3.017 | 0.00255 ** |
| Concordance= 0.599  (se = 0.029 ) | | | | | | |
| Likelihood ratio test= 9.41,   p=0.002 | | | | | | |

**b.**

| coxph(formula = Surv(time, event) ~ age + stage + groups) | | | | | | |
|---|---|---|---|---|---|---|
| | Coeff | Exp (Coeff) | Lower 0.95 | Upper 0.95 | z | Pr (>\|z\|) |
| Age | 0.0414 | 1.0423 | 1.0226 | 1.0625 | 4.252 | 2.12e-05 *** |
| Stage | 0.8405 | 2.3176 | 1.7821 | 3.0141 | 6.270 | 3.62e-10 *** |
| Groups (Lower) | -0.7093 | 0.4919 | 0.3098 | 0.7813 | -3.005 | 0.00265 ** |
| Concordance= 0.745  (se = 0.035 ) | | | | | | |
| Likelihood ratio test= 62.27,   p=2e-13 | | | | | | |

**Supplementary Table 1** Cox Proportional hazards model using a) Univariate Cox Regression b) Multivariate Cox Regression using age, stage of cancer and the groups calculated based on the generated scores for TCGA-CRC cohort.

**a.**

| coxph(formula = Surv(time, event) ~ groups) | | | | | | |
|---|---|---|---|---|---|---|
| | Coeff | Exp (Coeff) | Lower 0.95 | Upper 0.95 | z | Pr (>\|z\|) |
| Groups (Lower) | -0.4541 | 0.635 | 0.4541 | 0.888 | -2.654 | 0.00795 ** |
| Concordance= 0.558  (se = 0.023) | | | | | | |
| Likelihood ratio test=7.16,  p=0.007 | | | | | | |

**b.**

| coxph(formula = Surv(time, event) ~ age + stage + groups) | | | | | | |
|---|---|---|---|---|---|---|
| | Coeff | Exp (Coeff) | Lower 0.95 | Upper 0.95 | z | Pr (>\|z\|) |
| Age | 0.0339 | 1.0345 | 1.0156 | 1.054 | 3.600 | 0.000318 *** |
| Stage | 0.6586 | 1.9321 | 1.5074 | 2.476 | 5.201 | 1.98e-07 *** |
| Groups (Lower) | -0.3089 | 0.7342 | 0.5093 | 1.058 | -1.656 | 0.097811 . |
| Concordance= 0.668  (se = 0.027 ) | | | | | | |
| Likelihood ratio test= 42.12 ,  p=4e-09 | | | | | | |

**Supplementary Table 2** Cox Proportional hazards model using a) Univariate Cox Regression b) Multivariate Cox Regression using age, stage of cancer and the groups calculated based on the generated scores for TCGA-STAD cohort.

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
