# OpenReview forum: "Deep Learning for interpretable end-to-end survival prediction in gastrointestinal cancer histopathology"
_MICCAI.org/2021/Workshop/COMPAY — COMPAY 2021_

### Official Review · Reviewer_QocZ · 2021-08-02
**he paper presents the use of an end-to-end model that analyzes colorectal and gastrointestinal WSIs, to predict a risk score used to stratify them into high v. low risk of death.**

**Rating:** 5
**Confidence:** 4

**Review:**

**Strengths:**

The model is able to significantly stratify patients into two risk groups, shown using KM plots and log-rank tests. The model uses a common tiling approach to make it computationally feasible on a GPU. Moreover, both the images (TCGA) and the code is publicly available. The language of the text is clear and understandable.

**Weaknesses:**

The methodology does not seem original or novel, and using a Cox loss function had been done in many studies prior. In addition, the results are not convincingly showing that their model actually adds much predictive power over simply using the clinical variables age and cancer stage. Furthermore, there are rather many details missing in the methodology that need to be included.

**Questions for the rebuttal:**

1.	Hazard function does not necessarily have to be a baseline*risk function. You are talking about Cox Proportional Hazards model here it seems. Please clarify.
2.	You randomly selected 250 tiles per WSI. Why this number and how does that affect results? See the next point 3 which is a continuation of this.
3.	TCGA is known to have high tumor purity slides. How does this affect the applicability of your model on real-world data? Should be discussed.
4.	ResNet50 has 2048 features by default. You list 1024. Please explain your modification.
5.	Why ResNet50? Did you try other backbone models than ResNet50? How to they compare? Only very briefly mentioned in the discussion.
6.	Explain in the text how you aggregate tile predictions to WSI level. Mean, min, max..?
7.	Motivate your inclusion of stage and age in the CPH model. Why those, and only those variables? References needed.
8.	Supplementary table 1 A and B (TCGA-CRC) show that the c-index of using only the DL score is about 0.6, not much above random, and including age and stage also bumps it to 0.75. What is the c-index of using only stage and age? Similar or maybe even better than using only DL score? Same argument for table 2 (TCGA-STAD).
9.	The discussion mentions detection of important features, namely infiltrating lymphocytes, although there have been no experiments presented to show this or explanation of involvement of a pathologist for review.
10.	The methodology using a CPH loss is not novel. There seem to be relevant references omitted, e.g. DeepSurv (Katzmann, 2016) and EPIC-Survival (Muhammad, 2021).

**Minor comments:**

1.	Do not capitalize “Deep Learning”. Use “deep learning”.
2.	The abbreviated name “EE-Surv” needs to be explained at first use.
3.	“…survival Function S(t)…” should be “…survival function S(t)…”.
4.	“Hazard function…" should be “The hazard function…”.
5.	“…happend)" should be “…happened)”.
6.	In supp. Fig 2 B and C, explain which is which of rows and columns, high score tiles or high score patients.

---

### Official Review · Reviewer_Hve5 · 2021-08-19
**Review by Hve5**

**Rating:** 6
**Confidence:** 3

**Review:**

The paper proposes the prediction of survival directly from WSIs using two public datasets for colorectal cancer (TCGA-CRC) and gastric cancer (TCGA-STAD), using a pretrained ResNet backbone and a Cox proportional Hazard loss function. While the approach seems promising and there are commendable efforts towards establishing the statistical significance of the results obtained (something that a lot of other work in the field would do well to pursue in their articles!), the following points should ideally be considered by the authors:

1. The ResNet backbone used is pretrained on ImageNet, a dataset of natural images. I wonder if that is a suitable course of action, given the significant morphological differences between natural images and H&E images. Would it not be a better idea to pretrain on a histology dataset?
2. The survival score predicted from the H&E slides could potentially be linked to features at the slide level - some commentary on this, along with possible biological insights about such correlations, would be highly desirable.
3. A detailed description of the modelling process along with the preprocessing process in brief (while detailed in Kather et al., it may help to paraphrase some salient points) would be appreciated.

---

### Decision · Program_Chairs · 2021-08-25

Accept